Quantum readout and gradient deep learning model for secure and sustainable data access in IWSN

Alzubi Omar A. o.zubi@bau.edu.jo
Prince Abdullah Bin Ghazi Faculty of Information and Communication Technology, Al-Balqa Applied University , Al-Salt , Jordan
Aljawarneh Shadi
Electronic publication date: 2022 Jun 6
Publication date: 2022
Volume: 8
Electronic Location ID: e983
Received 2021 Aug 23; Accepted 2022 Apr 25
Copyright: ©2022 Alzubi
Copyright year: 2022
Copyright holder: Alzubi
License: This is an open access article distributed under the terms of the Creative Commons Attribution License, which permits unrestricted use, distribution, reproduction and adaptation in any medium and for any purpose provided that it is properly attributed. For attribution, the original author(s), title, publication source (PeerJ Computer Science) and either DOI or URL of the article must be cited.
License URL: https://creativecommons.org/licenses/by/4.0/

Keywords: Security, Machine learning, Deep learning, Data privacy, Industrial wireless sensor network, Quantum readout, Authentication, Authorization, Energy consumption

Funding: The Deanship of Scientific Research and Innovation at Al-Balqa Applied University, Al-Salt, Jordan DSR-2021#380 The research reported in this publication was funded by the Deanship of Scientific Research and Innovation at Al-Balqa Applied University, Al-Salt, Jordan. (Grant Number: DSR-2021#380. The funders had no role in study design, data collection and analysis, decision to publish, or preparation of the manuscript.

==============================
The industrial wireless sensor network (IWSN) is a surface-type of wireless sensor network (WSN) that suffers from high levels of security breaches and energy consumption. In modern complex industrial plants, it is essential to maintain the security, energy efficiency, and green sustainability of the network. In an IWSN, sensors are connected to the Internet in a non-monitored environment. Hence, non-authorized sensors can retrieve information from the IWSN. Therefore, to ensure that data access between sensors remains sustainable and secure, energy-efficient authentication and authorization are required. In this article, a novel Quantum Readout Gradient Secured Deep Learning (QR-GSDL) model is proposed to ensure that only trustworthy sensors can access IWSN data. The major objective of this QR-GSDL model is to create secure, energy-efficient IWSN to attain green sustainability and reduce the industrial impact on the environment. First, using the quantum readout and hash function, a registration method is designed to efficiently perform the registration process. Next, a gradient secured deep learning method is adopted to implement the authentication and authorization process in order to ensure energy-saving and secure data access. Simulations are conducted to evaluate the QR-GSDL model and compare its performance with that of three well-known models: online threshold anomaly detection, machine learning-based anomaly detection, and dynamic CNN. The simulation outcomes show that the proposed model is secure and energy-efficient for use in the IWSN. Moreover, the experimental results prove that the QR-SGDL model outperforms the existing models in terms of energy consumption, authentication rate, authentication time, and false acceptance rate.

Introduction

The underlying rationale for the recent conceptualization of the Industrial Internet of Things (IIoT) has been to leverage the Internet of Things (IoT) and apply its advantages to the industrial wireless sensor networks (IWSNs) in order to create interconnected industrial environments. IWSNs play an essential part in the management and operation of industrial machinery across a wide range of sectors. The main task of an IWSN is to monitor the performance of different devices through the collection, storage, and retrieval of data in real-time in an industrial environment. The application of the IWSNs framework in such systems is intended to increase optimization and improve industrial automation processes. Regardless of its advantages, the IWSN suffers from many security and privacy issues like data leaking, node compromise, authentication and authorization problems, data loss, and many more. The authentication problem is the most widespread concern among all the issues because all the sensing devices are located and accessed remotely. A false authentication may hamper the complete security of the system. The first security step must be strong so that no unauthorized user can access the system. Several authenticated key agreement schemes are proposed, but they are limited to only one device at a time. In some scenarios, the IIoT networks are closed so that no unauthorized user can access the internal networks of the system. However, the authentication problem for insider users is still present. An insider attack can be possible and creates an issue if the authentication mechanism is not strong. Thus, these issues motivated us to design a dynamic deep learning-based solution named the Quantum Readout Gradient secured Deep Learning (QR-GSDL) for the authenticity of the sensor-based IIoT networks. The proposed model will work on a different layer for insider user authentication and authorization purpose. The proposed authentication steps included an improvement in the authentication rate with minimum time consumption and delay.

One of the most well-known models that has been utilized in the Industrial Internet of Things (IIoT) domain is the online threshold anomaly detection model. It employs a learning method based on statistical formulations to distinguish the characteristics of devices and flag any differences in those characteristics as anomalies (Li et al., 2019). The model is independent in terms of device operations because statistical data about the system is acquired by using the IoT application program interface. For this model, multiple machine learning techniques have been introduced in the training process, and the performance on normal systems was designed in a similar manner. The model is able to detect anomalous activities in an efficient manner by summing cumulative operations and using localized outliers, thereby improving accuracy and simultaneously reducing false alarms. However, despite improvements in the accuracy and false alarm rates, this model does not address the issue of the security of data communication in the context of the industrial sector.

The other well-recognized model that has been employed in the IIoT is the machine learning-based anomaly detection model  (Zolanvari et al., 2019). This model was developed to address the most prevalent susceptibility in the IIoT, namely, the injection attack. There are three main forms of injection attacks that can be mitigated by applying machine learning techniques: command injection, structured query language, and backdoor attacks. Through the adoption of a machine learning-based approach, not only was it demonstrated that the attack detection rate was improved, but there was also a steep reduction in the value of the mean absolute error. However, despite the improvement in the attack detection rate and the minimization of the mean absolute error, the false-negative rate was not minimized by machine learning-based anomaly detection. In other words, anomaly intruders were still incorrectly detected, thus leading to a lack of overall efficacy.

In Yuan et al. (2020), a dynamic CNN (DCNN) technique is planned to learn the hierarchical local nonlinear dynamic features of soft sensor modeling. Every 1D process sample in DCNN is dynamically increased into a 2D data sample with lagged unlabeled process variables, comprising both spatial cross-relationships and temporal auto-correlations. Then, to derive the local nonlinear spatial-temporal function from the 2D sample data matrix, the convolutional and pooling layers are alternately used. In addition, the concept of how the local nonlinear spatial-temporal function can be taught from the network is studied for DCNN. In an industrial hydrocracking process, the efficacy of the proposed DCNN is tested. However, the authors did not provide any proof of the energy efficiency of their approach (Alzubi et al., 2020b; Alzubi et al., 2020a).

In this article, a deep learning-based solution is presented to overcome the security issues that currently exist in the authentication and authorization protocol for the industrial wireless sensor network (IWSN). The proposed solution employs a novel model named the Quantum Readout Gradient secured Deep Learning (QR-GSDL) model. This model first verifies the authenticity of a given sensor seeking access to data in the IWSN by using a quantum readout and hash (QRH) function. This registration process facilitates effective validation and therefore reduces the false acceptance rate. Next, the security issues inherent in the authentication and authorization procedure are addressed by using a gradient sparse auto deep learning algorithm. This type of algorithm was adopted because it was envisaged that its usage would lead to an improvement in the authentication rate (AR) with minimum time consumption and delay. Accordingly, the designed model substantially minimizes the false acceptance rate (FAR), leading to an improvement in both the authentication rate and authentication time (AT).

We believe that the best-suited real-world environment to implement our proposed QR-GSDL model is in industrial applications such as machine health, automated metering, remote monitoring, and staff management. The only requirement is that the pre-defined setting of IWSN be stationary.

Related Work

With the advancement of technologies, because of their benefits over conventional wired networks, wireless sensor networks (WSNs) have fantastic deployment opportunities for industrial scenarios. However, fully integrated mechanized processes and wireless networking conditions allow the high security and low energy consumption requirements of industrial wireless sensor networks (IWSNs) more stringent. We will discuss the relevant work in this section from the point of view of security and energy consumption. Many researchers presume industrial wireless sensor networks and present different authentication and authorization schemes. However, these schemes were not ideal for IWSN. This is due to the fact that in terms of energy efficiency and computing overhead, node authentication by cluster head on a regular basis results in considerable overhead.

Several studies have been conducted in the area of deep learning to make the IoT-enabled WSN more efficient, robust, and secure (Alzubi et al., 2019b). The works that are most relevant to this paper include the deep learning model that was proposed in Liang et al. (2020). This model is based on edge computing and aimed at minimizing the traffic (data transmissions) in the network to reduce network congestion while maintaining classification accuracy. However, the method in Liang et al. (2020) did not provide each user with data privacy. Therefore, to address this privacy issue, two privacy-preserving deep learning models named DeepPAR and DeepDPA were presented in Zhang et al. (2020). The DeepPAR model offered a mechanism that prevented a user’s information from being leaked to others while keeping the secrecy level dynamically updated. To address this issue, the DeepDPA model applied a set of key management techniques to guarantee the backward secrecy of group participants. However, DeepDPA and DeepPAR were not able to minimize the false acceptance rate. Therefore, the design of the proposed QR-GSDL model is aimed at addressing the data privacy problem while at the same time minimizing the false acceptance rate.

Another deep learning model was proposed in Liao et al. (2019) in order to improve the authentication performance of the IWSN. The model employed three methods to authenticate sensor nodes. Each of these methods was based on a machine learning algorithm. The first one applied an improved algorithm based on a convolution preprocessing neural network (CPNN), the second utilized a deep neural network, and the third one used a convolutional neural network. Although the model required minimal computing resources to reduce the latency in performing multi-node authentication, it failed to reduce the authentication time, even when using a so-called improved CPNN-based algorithm. Hence, it is anticipated that the proposed QR-GSDL model will overcome the time consumption-related shortcoming encountered in Liao et al. (2019) through the use of a quantum readout and hash (QRH) function to verify and validate the authentication of the sensor in a minimal amount of time. It is also envisaged that the use of a QRH function in the proposed QR-GSDL model will also be able to minimize the volume of network traffic and consequently reduce communication costs. Thus, overcoming the communication cost limitation of the deployment-based optimization model that was introduced in Li et al. (2017) to ensure network security and simultaneously improve network lifetime.

Despite the above achievements in the area of deep learning, a survey of the application of deep learning tools in the smart industry presented in Ma et al. (2019) concluded that while deep learning provides an opportunity to solve many classical issues, authentication and authorization problems are not tackled. Therefore, as a first step in addressing these problems, the QR-GSDL model is designed in such a way as to ensure that the gateway node in the IWSN checks the authenticity of the sensor node that is seeking access to information in the IWSN, thereby guaranteeing correct and appropriate authorization.

Other works have also explored methods to improve authentication. For instance, in Chen, Lee & Lin (2020), a secure authentication scheme was introduced that depended on credential and dynamic IDs for WSNs in IoT environments. For the scheme, an authentication key agreement protocol based on three parties was designed using the Burrows–Abhadi–Needham logic method. It was reported that the scheme was able to ensure low computational and communication costs, but it was admitted that the false acceptance rate was not improved. Consequently, the gradient secured deep learning method is integrated into the proposed QR-GSDL model in order to achieve a reduction in the false acceptance rate.

Alrabea, Alzubi & Alzubi (2020) and Alzubi et al. (2019a) propose a re-authentication scheme for the Voronoi graph-based network model. The scheme maintains anonymity while using fewer resources than the previous schemes. The system, however, suggests neighbor wandering, which might not be ideal for a realistic situation. Also, they did prove the efficiency of their model in terms of energy consumption (Alrabea, Alzubi & Alzubi, 2020; Alzubi et al., 2019a).

In an alternative attempt to improve the security aspect of the WSN, a convolutional technique (CT) was developed in Alghamdi (2019), which involved generating security bits using convolutional codes. The aim of the CT was to protect the WSN from attacks caused by malicious nodes. The designed technique improved network security and minimized computational complexity because no key distribution was needed. However, authentication time was not minimized by CT. Thus, a gradient secured deep learning method is included in the proposed QR-GSDL model in order to attempt to reduce the time consumed in the authentication procedure.

Industrial wireless sensor networks, which are an evolved category of WSN in which sensors are combined to monitor the status of equipment and to control systems in real-time, also have limitations related to security, privacy, and energy. To address these drawbacks in the IWSN context, a lightweight decision-making framework based on trust value identification was designed in Ramesh & Yaashuwanth (2019). The lightweight trust framework was used for quality of service clustering in order to perform the secure routing process. A quantifiable trust value was determined through the cluster head within the cluster. It was claimed that flawed, untrusted, counteract, and malicious nodes could be predicted using this framework. However, the communication cost was not minimized. Therefore, in the QR-GSDL model, quantum sparse auto-encoding and decoding are employed to reduce the communication cost in the IWSN system.

A different protocol named secure directed diffusion was suggested in Sengupta, Ruj & Das Bit (2018). This protocol depended on binding the node’s geographic location and ID in order to induce a cryptographic key based on the location. The produced key then formed the foundation of a neighborhood authentication process for the IWSN. However, only theoretical statements were provided regarding the effectiveness of this authentication process and the computational overhead.

On the other hand, in Qureshi et al. (2020), a centroid position analysis was performed in an attempt to decrease data transmission failure and delay. In addition, a gateway clustering routing protocol was used for cluster head selection from the centroid position. Then the gateway node minimized the load from the cluster head nodes and transmitted the data to the base station. However, security issues were not taken into consideration. Therefore, in our proposed approach, an authentication process is carried out to establish secure communication.

Cooperation between the sensors that are communicating with a central base station is one of the factors that contribute to security. In light of this, cryptographic algorithms based on secret keys were designed in Tahir, Tahir & McDonald-Maier (2018) in which the ICMetric method employed the device features to generate secret keys for use in cryptographic services. However, the proposed method failed to offer an effective authentication process that at the same time did not increase resource overheads. Hence, in the proposed QR-GSDL model, a gradient secured deep learning algorithm performs the authentication to allow secure data communication.

An energy-efficient data transmission mechanism is proposed to improve emergency data transmission by increasing accuracy and decreasing packet delay (Sheikh et al., 2012; Singanamalla et al., 2019; Nazir et al., 2020). It was claimed that the scheme was reliable, but a mechanism for ensuring the security of data transmission was not presented. In contrast, an authentication process is carried out in the QR-GSDL model to enable secure data transmission.

A cooperative mechanism was presented in Iqbal, Kim & Lee (2017) to reduce both the false alarm rate and energy consumption. The designed mechanism improved the probability of accurate decisions being made at a specified signal level. Also, the suggested mechanism in Iqbal, Kim & Lee (2017) was reported to be able to achieve a reduction in the false alarm rate, but only for indoor IWSNs. However, it was not particularly efficient in terms of energy consumption in relation to the time needed to perform its computations. Therefore, quantum sparse encoding and decoding are used in our proposed QR-GSDL model in order to reduce computation time (Sheikh, Ahmad & Fan, 2016; Alrabea, Alzubi & Alzubi, 2019; Alzubi et al., 2014).

The three issues of security, network lifetime, and coverage were handled in Cao et al. (2020) by converting the disjoint routing paths to address the flow issues. However, despite an improvement being observed in security and coverage, optimization and operation time were not focused on. Therefore, again the quantum sparse encoding and decoding approach is used in the proposed model in order to minimize time consumption.

To address the issue of security, in Cao et al. (2019), multi-objective evolutionary algorithms were designed for a heterogeneous WSN. Moreover, a 3D signal propagation model used the line-of-sight idea to determine the signal path loss. However, the security level was not improved by the designed model. Hence, the gradient secured deep learning model is employed in the current study to perform authentication for secure data communication.

Another password-based authentication scheme was proposed in Lee et al. (2018) to verify security with minimal communication and computation cost. However, the communication and computation overheads were not minimized by the developed password-based authentication scheme. Therefore, in the proposed model, quantum sparse encoding and decoding are used with the expectation that this approach can reduce the computation overhead.

Lastly, a mutual authentication system integrating temporal credentials and multiple passwords was proposed in Liu, Zhang & Liu (2017) in order to minimize the overheads. However, while the authentication time was reduced, the false alarm rate was not minimized. Therefore, in the proposed QR-SGDL model, the QRH is used to offload the false alarm rate.

Motivated by the above issues encountered in related studies, in this work, a novel model for IWSN, named the Quantum Readout Gradient Secured Deep Learning (QR-GSDL) model, is developed in order to improve not only the authentication rate but also the authentication time and false acceptance rate. In the following, an elaboration of the QR-GSDL model is presented.

Quantum readout gradient secured deep learning in IWSN

In this section, we present our proposed model, QR-SGDL, which consists of three phases: (i) secure and energy-efficient localization, (ii) sensor node registration, and (iii) authentication. The system model and the three phases of QR-SGDL are elaborated on in the following subsections.

System model

Figure 1 provides the system model of IWSN where the QR-SGDL model will be implemented. The IWSN system model comprises four entities: a number of sensor nodes, a number of base stations, a gateway node and a supervisory control unit connected to a network (Zolanvari et al., 2019).

Generally, several different sensors are deployed in an industrial plant for monitoring purposes. Here, the sensors which consume lesser energy for sensing the industrial plants are chosen. These sensors gather the data packets from their adjacent environment and communicate them to the gateway node via one or more base stations. Next, the gateway node transmits the acquired data packets to the supervisory control unit through a secure channel.

The gateway node gathered two types of packets—data packets and control packets—from the sensor nodes. Before collecting the data packets, the gateway node ensures that these packets are originated from authenticated sensor nodes and verifies whether the sensor node has been tampered with or not. This will be accomplished by applying the quantum readout hash registration which is explained thoroughly in algorithm 1. After attaining the data, the QR-GSDL method analyzes the data about industrial plants in order to maintain green sustainability.

Figure 1 System model for an industrial wireless sensor network.

System architecture of the QR-SGDP model

The proposed QR-SGDL model uses deep learning concepts to perform multiple processes in several layers. The deep learning network uses one input layer, two hidden layers, and one output layer for improving security during data access and to improve the green sustainability of the network. The feed-forward fashion deep learning network collects the nodes in the input layer, learns in the hidden layers, and transforms the results into an output layer.

The system architecture of the QR-SGDL model is designed to address energy and green sustainability issues by analyzing the industrial plant’s data. It also aims to handle security issues such as authentication and authorization in the IWSN by achieving a minimum false acceptance rate.

In the IWSN, the upper layer of the architecture is a transmission layer comprising base stations, sensor nodes, gateway node, and supervisory control unit. All of these elements can be found in any industry-enabled architecture, and this configuration realizes the separation of the deep learning model and the IWSN model. The energy-efficient sensor node represents the input part while the supervisory control node represents the calculation part. With the deep learning model enabled in the IWSN, we construct a quantum sparse auto encoder (QSAE). In the proposed model, the QSAE handles energy, green sustainability, and security issues (i.e., authentication and authorization) in the IWSN.

During the training stage, the supervisory control unit sends a data access request made by a given sensor to the input layers. The QSAE checks for energy availability, authenticity, and authorization of the sensor by employing the quantum readout function to ensure smooth communication between the sensor and the supervisory control unit with a minimum false acceptance rate. The verification of energy availability, authenticity, and authorization by QSAE will be discussed in the coming sections. The supervisory control unit thus ensures energy consumption, authenticity, and authorization via the gateway node based on this security mechanism. When the sensor has been authenticated and authorized, it is allowed to implement its process.

Sensor node registration phase

The registration process for the minimal energy consumed sensor is carried out by the gateway node by means of a QRH model. This QRH model is used because it can accurately verify the authenticity of an object. The registration procedure employed by the QRH model is illustrated by means of an activity diagram in Fig. 2.

Let us assume that a sensor S = (S1, S2, …, Sn) wants to access the data (or data packets) DP = (DP1, DP2, …, DPn) in IWSN. The sensor registers the sensor’s details in the gateway node GN. To minimize the false acceptance rate, an integrated QRH function is used. The pseudocode representation of Quantum Readout Hash registration is detailed in Algorithm 1.

Figure 2 Registration activity diagram.

_____________________________________________________________________________________ Algorithm 1 Quantum readout hash registration Input:  Sensor S  =  S1,S2,...,Sn,  Gateway Node GN,  Base Station BS  = BS1,BS2,...,BSn Output: Authentic sensor registration Begin:   for each Sensor S with Gateway Node GN and Base Station BS do       Obtain identity and request registration using equation 1        Obtain arbitrary challenge using equation 2        Obtain single quantum using equation 3        Obtain interim identity and quasi congruence using equation 4        Evaluate hash function using using equation 5        Return: authentic sensor registration   end for End _____________________________________________________________________________________

The sensor registration process explained in Algorithm 1, starts when sensor S chooses an identity SID and relays (SID, RReg) to the gateway node via a secure channel. Here, RReg represents the request made by the sensor for registration. (1) S:SID,RReg→GN.

Upon reception of the request made by the sensor for registration, the gateway node produces an arbitrary challenge CS for the sensor and sends CS to the sensor S via a secure channel as given by: (2) GN:CS→S.

After receiving CS, the sensor extracts the quantum readout QR outputs RS = QD(CS) and sends RS to the gateway node GN. Here, QR outputs RS to verify the authenticity of quantum data packet QD that claims to be from a given source or sensor. A single quantum Q is produced in a random manner for each sensor, therefore minimizing the false accept rate, the energy consumption, and improving the authenticity rate. (3) S:RS→GN.

Subsequently, the gateway node produces an interim identity IIDS and a set of Quasi Congruence QC = (qc1, qc2, …, qcn) and then transmits (IIDS, QC) to the sensor S in order to facilitate communication with S. (4) GN:IIDS,QC→S.

Finally, the sensor S input a one-time password OTP for the particular session s and stores them in its base station. Next, S extracts the hash intermediate output αs = QD(OTPs). The sensor selects the session password spwds and inputs spwds into the base station which evaluates the hash function β for validation, as presented in Eq. (5). (5) β=HASHαs||spwds.

By applying the above-integrated functions, it is considered to be possible to achieve authentic sensor registration with a minimum false acceptance rate.

Gradient secured deep learning model

Let us assume that a sensor S wants to acquire industrial plant data from a specific sensor in the IWSN. Then, mutual authentication between the two sensors Si and Sj has to be established, where the identity of the sensor has to be checked prior to providing access. With authenticated nodes, the gateway node checks sensor node access for guaranteeing energy-efficient authorization. First, authentication is performed by means of gradient secure localization method, and then energy-efficient authorization is done by means of a quantum sparse encoding and decoding approach. The activity diagram for the energy-efficient authentication and authorization procedure is shown in Fig. 3. More details are given for the proposed authentication process as a flow diagram, which is shown in Fig. 4. The pseudo code representation of Gradient Sparse Auto Deep Learning for energy-efficient and secure communication is detailed in Algorithm 2.

As presented in Algorithm 2, the mutual authentication process performed by the gateway node GN and the designated sensor is as follows. First, the sensor S initially inputs a one-time password OTPS Next, the base station extracts the quantum readout QR output RS. Then, the gateway node GN calculates β′ = HASH(αs||spwds) and compares the computed β′ with the stored β. In this work, the comparison is done based on the localization concept, whereby industrial data in the IIoT are gathered by sensors in distinct locations and sent to the servers to facilitate analysis. Most of the current localization systems measure local deviation with respect to neighbor based on reachability distance (Li et al., 2019) or according to local deviation factor. However, in the real world, those that are present within the IIoT environment may attempt to interrupt this localization process. When successful, such attacks may compromise certain sensors and thus critically falsify the entire environment. To address this issue, first, in this work, an energy-efficient and secure localization method based on gradient associating sensors is presented.

Figure 3 Energy-efficient authentication and authorization activity diagram.

Figure 4 Flow diagram of proposed authentication process.

__________________________________________________________________________________________ Algorithm 2 Gradient sparse auto deep learning Input:  Sensor S  =  S1,S2,...,Sn,  Gateway Node GN,  Base Station BS  = BS1,BS2,...,BSn Output: energy-efficient and secure communication Begin:   for each Sensor S with Gateway Node GN and Base Station BS do       Obtain energy-efficient secure localization based on gradients associating    sensors using equations 9 and 10        Authentication:       if disclosed locations equal then           Authentication successful            Go to Authorization       else           Authentication is not successful            Session is terminated        end if       Authorization:       Perform Quantum Sparse Auto-Encoding using equation 11        Perform Quantum Sparse Auto Decoding using equation 13        if R=Q then           energy-efficient authorization is successful            energy-efficient secure communication        else           energy-efficient authorization not successful            Session is terminated        end if   end for End _____________________________________________________________________________________

Let us consider sensors Si and Sj at different locations A1, B1, A2 and B2, mathematically formulated as given below. (6) Si0=A1+B1

(7) Sj0=A2+B2.

Based on Eqs. (6) and (7), the distance between the sensors is mathematically calculated by applying Eq. (8). (8) Sij=A1+A22+B1+B22.

However, in order to analyze the relation between gradient associating sensors, it is computed by Eqs. (9) and (10). (9) cosθ=Si0+Sj0−SiSj22Si0Q0

(10) cosθ=A1A2+B1B2A12+B12A22+B22.

By introducing the Gradient Secure Localization (GSL) method, security, energy efficiency, and green sustainability can be ensured even in the presence of attacks. In this manner, with the location verification by the GSL method, the precision or exactness of the disclosed locations of the sensors is made in an effective fashion, therefore ensuring authentication. Note that the comparison here is made based on the disclosed locations. If they are not equal, the session is terminated. Upon successful comparison, the base station perceives S as a normal sensor.

Next, the base station generates a nonce NS and selects IIDS as the interim identity of the sensor, and sends the signing message IIDS, NS via a secure channel. After obtaining the signing on message, the gateway node GN initially locates the IIDS in its database. The gateway node then chooses the QRCRP(CS, RS), generates the nonce NGN and determines N′GN = NGN⊕RS and criterion Zo = H(N′GN||RS||NS). With the obtained criterion, in order to ensure that only authorized sensors communicate with each other, encryption and decryption are conducted using the QSAE. Moreover, the presence of three different layers in the system architecture (i.e., input layer, hidden layer, and output layer) reduces the false acceptance rate and ensures more accurate energy-efficient authorization. Here, R represents the feature expression of the output layer and encoding is performed as shown in Eq. (11). (11) R=δW∗DP+b.

From Eq. (11), the data packets required by the sensor are represented as vector DP = [DP1, DP2, …, DPn], where n denotes the total number of the input sensors. In addition, vector R = [R1, R2, …, Rm], represents the feature expression of the hidden layer, where m represents the sensors of the hidden layers. Finally, b represents the bias vectors and W the weight matrix from input to hidden layer, respectively, while δ denotes the activation function. Finally, the gateway node formulates a message and sends it to the respective sensor as presented in Eq. (12). (12) N:CS,N′GN,Zo,R→S.

Upon reception of the message as given 12, the base station BS of the corresponding sensor S extracts RS = QD(CS) and verifies the criterion Zo. Upon successful verification, the base station asks S to input its identity SiID and the sensor identity SjID that it needs to access. The decoding process obtains the reconstructed vector R of the output layer from the hidden layer value R, and this is mathematically formulated as in Eq. (13). (13) Q=gR=δWTR+b.

In Eq. (13), Q = [Q(1), Q(2), …, Q(n)] and WT represents the weight. Upon reception of the decoding output Q, the gateway node establishes the output. If the verification is successful, then the gateway node authorizes the sensor, and successful energy-efficient communication is thus established between the sensors Si and Sj; otherwise, the process is terminated.

Simulation setup

In order to examine the performance of the proposed QR-GSDL model in an IIoT system operation, a simulation environment is set up using Network Simulator 2 (NS2) with plants data collected by IoT devices downloaded from Kaggle (Kaggle, 2020). The dataset is composed of seven attributes and 16,382 instances. The attributes are demand_response, area, season, energy, cost, pair no, and distance. The dataset comprises the common information to facilitate the development of a demand response (DR) energy management system for industrial customers. The IoT platform improves the inter connectivity of entities in industrial energy management systems and minimizes the energy costs of industrial facilities. In our simulations, networks with a designated number of sensors are distributed in a random pattern within an area of 1500 m times 1500 m. The number of sensors is varied from 50 to 500. The positioning of nodes is made in a random fashion. Finally, the chosen simulation runs were 10 due to the fact that after the 10th run, there was a very small gain in the criteria values. We believe this is an advantage of our proposed model where it converges after a low number of simulation runs compared with the existing models in the literature (Li et al., 2019; Zolanvari et al., 2019; Yuan et al., 2020) where they converge after 45 simulation runs. Table 1 provides the simulation parameters used in our work.

Table 1 Simulation parameters.

Parameters	Description	
Simulation time	50 s	
Area size	1,500 m ×1,500 m	
Number of sensors	50, 100, 150, 200, 250, 300, 350, 400, 450, 500	
Sensor placement	Random distribution	
Transmission range	400 m	
Simulation runs	10	

Results and Discussion

The performance of the proposed QR-GSDL is compared with three well-known models: online threshold anomaly detection (Li et al., 2019), machine learning-based anomaly detection (Zolanvari et al., 2019), and dynamic CNN (Yuan et al., 2020). The performance analysis is based on three measures: energy consumption, false acceptance rate, authentication rate, and authentication time. The experimental results are presented in the form of tables and graphs.

Performance analysis of energy consumption

Energy consumption is defined as the product of a number of samples and energy consumed by one sensor node for performing authorization to achieve secured communication green sustainability. It is computed as:

Energy consumption is defined as the product of a number of samples and energy consumed by one sensor node for performing authorization to achieve secured communication green sustainability. It is computed as: (14) EC= ∑i=1nSamplesi×Energyconsumedbyonesensornode.

In Eq. (14), the energy consumption EC is computed based on the samples considered for experimentation (Samplesi) and energy consumed for authorization to achieve secured communication and green sustainability. It is measured in terms of joules (J). The energy consumed for four different methods is given in Table 2.

Table 2 and Fig. 5 describe the energy consumption of the proposed method compared with the three existing methods for a different number of sensors. The attained results illustrate that When the number of sensors increases, the energy consumption by the sensor during the authorization also gets increased linearly. However the the energy consumption of QR-GSDL model is lesser when compared to Li et al., (2019), Zolanvari et al. (2019) and Yuan et al., (2020). The sample simulations carried out with 50 sensors show that the amount of energy consumed by one sensor for authorization using QR-GSDL is 0.26J while energy consumed by one sensor using Li et al. (2019) is 0.36J, using Zolanvari et al. (2019) is 0.46J, and using Yuan et al. (2020) is 0.50J. The energy consumption saving is due to the application of the Gradient Sparse Auto Deep Learning algorithm where the gateway node checks data access by the sensors for ensuring energy-efficient and green sustainability. It is clear from the obtained results that QR-GSDL reduces the energy consumption by 13%, 20%, and 28% when compared with Li et al. (2019), Zolanvari et al. (2019) and Yuan et al., (2020), respectively.

Table 2 Energy consumption for different models and sensors.

Number of sensors	False acceptance rate	
	QR-GSDL	Online threshold
anomaly detection	Machine learning-based
anomaly detection	Dynamic
CNN	
50	13	18	23	25	
100	21	28	35	39	
150	34	40	46	50	
200	49	59	64	67	
250	63	70	73	76	
300	76	85	89	90	
350	94	101	105	108	
400	107	115	118	120	
450	118	126	129	132	
500	134	142	145	149	

Figure 5 Comparisons of energy consumption.

Performance analysis of false acceptance rate

The false acceptance rate (FAR) is a measure of the likelihood that the IWSN will incorrectly accept an access attempt made by an unauthorized sensor. The false acceptance rate is formulated as the percentage ratio of the number of false acceptances (FA) to the number of sensors (Samples) as input, as in Eq. (15): (15) FAR=FASamples100.

In Eq. (15), FA denotes the false acceptance made and a number of samples (sensors) Samples provided as input. It is computed in terms of percentage (%). The results of the false acceptance rate in modeling green sustainability issues in IWSN are summarized in Table 3.

The false acceptance rates generated by the IWSN when using the proposed method and the two compared methods are presented in Table 3 and graphically illustrated in Fig. 6.

It can be seen from Table 3 and Fig. 6 that when the number of sensors increases, the number of sensors that are checked for authenticity by the supervisory control unit via the gateway node also increased. Correspondingly, in all four models, the false acceptance rate increases. The false rate cannot be optimized by the QR-GSDL model.

However, the proposed model is able to reduce the rate when compared to the three models. As an example, when there are 150 sensors in the simulation, the number of sensors whose information is incorrectly accepted for transmission is 15 using QR-GSDL compared to 28, 45, and 48 using the online threshold anomaly detection (Li et al., 2019), machine learning-based anomaly detection (Zolanvari et al., 2019), and dynamic CNN (Yuan et al., 2020), respectively. Therefore, it can be inferred that the false acceptance rate is improved by QR-GSDL. It is considered that this is due to the application of an integration function, namely, QRH, which verifies and validates the authentication of the corresponding sensor in an effective manner. Also, the application of interim identity and quasi congruence results in the generation of distinct and unique identities that are not stored in the gateway node but held by the supervisory control unit. Hence the level of complexity and the false acceptance rate is reduced using QR-GSDL by 39%, 56%, 58% compared to Li et al. (2019), Zolanvari et al. (2019) and Yuan et al. (2020), respectively.

Table 3 False acceptance rate for different models and sensors.

Number of sensors	False acceptance rate	
	QR-GSDL	Online threshold
anomaly detection	Machine learning-based
anomaly detection	Dynamic
CNN	
50	8	10	12	13	
100	12	15	25	23	
150	15	28	45	48	
200	25	45	60	50	
250	30	50	75	65	
300	30	58	80	68	
350	25	40	85	79	
400	40	65	70	73	
450	45	80	95	87	
500	40	75	100	82	

Figure 6 Comparisons of false acceptance rate.

Performance analysis of authentication rate

Authentication rate (AR) is calculated as the percentage ratio of successful authentications (SA) made by the gateway node to the number of sensors (Samples), as in Eq. (16): (16) AR=SASamples100.

The authentication rates produced by the IWSN for different numbers of sensors (ranging from 50 to 500) when using the three tested methods are presented in Table 4 and Fig. 7.

As the number of sensors increased, the authentication rate for all four models also increased. This is because as the number of sensors increases, the frequency of sensors in the gateway node via the base station increases, so there is a higher probability of a longer amount of time being consumed for encryption and decryption to deal with the request made by each sensor.

However, significant improvement and gain increasing are trends that can be observed with the QR-GSDL approach. For instance, in the case of the simulation using 50 sensors, a total of 46 sensors are correctly authenticated as authentic sensors by the gateway node when applying QR-GSDL, whereas only 44, 41, and 40 sensors are correctly identified by applying (Li et al., 2019; Zolanvari et al., 2019), and (Yuan et al., 2020), respectively. Thus the authentication rate is higher with QR-GSDL compared to Li et al. (2019), Zolanvari et al. (2019) and Yuan et al. (2020). It is considered that this improvement is due to the application of a gradient sparse auto deep learning algorithm. By applying this algorithm, localization is first achieved using gradients for each sensor in a secure manner, rather than just by identifying the distance between the sensors based on their neighbors (Li et al., 2019). Hence the proposed method leads to a higher rate of correct authentications being made by the supervisory control via the gateway node of 5%, 8%, and 9% compared to Li et al. (2019), Zolanvari et al. (2019) and Yuan et al. (2020), respectively.

Table 4 Authentication rate for different models and sensors.

Number of sensors	False acceptance rate	
	QR-GSDL	Online threshold
anomaly detection	Machine learning-based
anomaly detection	Dynamic
CNN	
50	92	88	82	80	
100	90	86	80	78	
150	85	81	78	74	
200	88	83	75	70	
250	86	82	74	68	
300	88	84	75	70	
350	90	85	78	74	
400	85	81	75	71	
450	87	82	77	73	
500	89	85	79	75	

Figure 7 Comparisons of authentication rate.

Performance analysis of authentication time

Authentication time refers to the time consumed in authenticating the sensors as either normal or abnormal (malicious). The mathematical formula used to compute authentication time is shown in Eq. (17): (17) AT= ∑i=1nSamplesi×TimeR+Q

where AT is authentication time that is measured based on the samples considered for experimentation (Samplesi) and the time consumed in encryption (R) and decryption (Q). The value is given in milliseconds (ms). Table 5 and Fig. 8 present authentication times of the different models for different numbers of sensors.

Table 5 and Fig. 8 show that a significant improvement in authentication time is achieved by QR-GSDL as compared to Li et al. (2019), Zolanvari et al. (2019) and Yuan et al. (2020). As an example, in the case of the simulation conducted on 50 sensors, a total authentication time of 28.5 ms was required by QR-GSDL, compared to 35.5 ms, 43 ms, and 47 ms in the case of Li et al. (2019), Zolanvari et al. (2019) and Yuan et al. (2020), respectively. It is considered that this gain is due to the application of the gradient sparse auto deep learning algorithm. Furthermore, the authentication time consumption of the QR-GSDL is 13%, 23%, and 27% lower than that of the online threshold anomaly detection (Li et al., 2019), machine learning-based anomaly detection (Zolanvari et al., 2019), and dynamic CNN (Yuan et al., 2020), respectively.

Table 5 Authentication time for different models and sensors.

Number
of sensors	False acceptance rate	
	QR-GSDL	Online threshold
anomaly detection	Machine learning-based
anomaly detection	Dynamic
CNN	
50	28.50	35.50	43	47	
100	42.40	55.25	65.35	69.75	
150	65.25	70.35	80.25	85.35	
200	83.50	100.25	125.55	130	
250	105.25	125.35	135.35	138.45	
300	125.35	140.55	145.55	150.25	
350	145.55	175.35	180.25	186.75	
400	170	190	200.35	220.15	
450	182.35	195.25	225.55	238.15	
500	190	200.35	245.55	250.55	

Figure 8 Comparisons of authentication time.

It can be summarized that the proposed QR-GSDL outperforms the existing methods for the following three reasons: (i) Authenticity of the sensors is performed by the supervisory control via the gateway node by using gradients. (ii) Authorization of the authenticated sensors to access the data transmitted between them is carried out by using quantum sparse auto- encoding. (iii) Intermediate calculations are made at the supervisory control level and not at the gateway node.

Conclusion

This article formalized the authentication and authorization problems in IWSN. The proposed security protocol concentrated on the authenticity and authorization of the massive amount of data interchanged between the sensors. We proposed a Quantum Readout Gradient Secured Deep Learning method to improve authentication rate and time with a minimum false acceptance rate based on our problem formalization. The proposed authentication protocol performs several operations on different layers. The deep learning model uses one input layer, two hidden layers, and one output layer to improve the security model. This protocol first verifies the authenticity of the sensors that want to access the IWSN network data. This was performed by integrating quantum readout and hash functions. To deploy the proposed QR-GSDL method, the location information of the sensors is extracted to reduce the false acceptance rate of the method. The proposed model also includes a minimal energy-based registration process that correctly verifies the login process of the sensors. The Quantum Sparse Auto-Encoding and Decoding model ensures mutual authentication and authorization between the sensor nodes. A comprehensive simulation environment was designed using Network Simulator 2 with different sensors that were run ten times in the simulation with an area size of 1,500 m × 1,500 m for the implementation of the QR-GSDL authentication process. The performance of the proposed model is evaluated on the plants’ data and compared with the other three similar types of work. The performance comparison is based on the energy consumption, false acceptance rate, authentication rate, and authentication time. The achieved experimental results indicate that our proposed deep learning-based Sparse Auto-Encoding and Decoding model not only ensures better authentication but also better authorization compared to the state-of-the-art methods.

Future Work

Application of the proposed model in the field of the Internet of Things (IoT) is left as future work. It will be interesting to apply some adaptions to the QR-GSDL model and perform experiments to evaluate its performance in the IoT. In addition, it could be interesting to deeply analyze the performance of the QR-GSDL model on different datasets.

Supplemental Information

Supplemental Information 1 The code to generate the first experiment

Click here for additional data file.

Supplemental Information 2 The code to generate the second experiment

Click here for additional data file.

Supplemental Information 3 The code to generate the third experiment

Click here for additional data file.

Supplemental Information 4 Raw data

Click here for additional data file.

Additional Information and Declarations

Competing Interests

Author Contributions

Data Availability

The author declares that there are no competing interests.

Omar A. Alzubi conceived and designed the experiments, performed the experiments, analyzed the data, performed the computation work, prepared figures and/or tables, authored or reviewed drafts of the article, and approved the final draft.

The following information was supplied regarding data availability:

The NS2 codes and the raw data are available in the Supplementary Files.

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
