# Peer review of "Quantum readout and gradient deep learning model for secure and sustainable data access in IWSN"

_PeerJ Computer Science, doi:10.7717/peerj-cs.983_

## Round 0.1 · original submission · Minor Revisions

The author should follow the comments from the reviewers

·

Basic reporting

1.Use of deep learning models with wsn is good for intrusion detection


2. For authentication and authorisation alone ML/DL are not required, however to minimise the false acceptance rate use of sophisticated algorithms is good technique

Experimental design

3. It is generally preferable to use light weight techniques in wsn/ IoT systems , deep learning is time and resource intensive when it comes to training. If a certain protocol has to be followed for authentication ,violation of it leads to an alarm. To detect this violation an anomaly based model is used . The same can be implemented without a model by simply using a checklist that contains a exhaustive list of items that need to be covered before the device can be safely added to the network. So while the approach of the author seems to be effective for now , better algorithms will keep coming up. To setup an authentication based on checklist mechanism is however time resilient unlike a learning based method. Statistical analysis of data could be useful from time to time , but for wsn lesser data that needs to be processed the better.



4. Deep learning and machine learning models tend to work based on historic data and behaviour . Training the models when planned changes occur in the infrastructure is an operational overhead.

Validity of the findings

5 gradient based algorithm is not the best approach in security mechanisms as , things need to be as deterministic as possible . Sometimes the position of a sensor or its configuration could be constantly changing in the IWSN . In such a case it's not clear how this system works.

Additional comments

No comments

Reviewer 2 ·

Basic reporting

The English language of the paper is clear and readable.

• There is a need to explain the authentication problem in more details, some of the IIoT networks are designed in a closed network where they have their own cables and devices and no one outside the network has any access to it, in this case the authentication problem needs to be clarified at what circumstances it is considered an issue. I suggest the authors provide an example to explain when/how the authentication step should be included in the design.

• Line 67: “in this paper” is repeated.

• Authors need to identify the terminologies earlier in the paper such as authentication rate, authentication acceptance false rate, and authentication time.

• Figure 1 in the paper should me improved to tell more details about the system model; not much information can be extracted from the figure in its current design.

• Line 213: “the gateway node has to approve the authenticity of the sensors. The gateway node verifies whether the sensor has been tampered with or not in order to guarantee the confidentiality and integrity of the data packets”. Authors are required to identify what the gateway node is, what type of data is gathered, and how the authentication, confidentiality, and integrity of the sensors are being verified?

• Line 220: Extra word “input”

• Figure 2 is meaningless and not clear, E.g., What is the input look like? Etc.,.

• The paper has no example of where their design is applicable.

• Line 237, “The QSAE checks for energy availability, authenticity and authorization of the sensor by employing the quantum readout function to ensure smooth communication between the sensor and the supervisory control unit with a minimum false acceptance rate”. How are the energy availability, authenticity, and authorization verified by the QSAE? Authors should mention that it will be explained in coming sections.

• QRH should be used in full words for the first time then used as a short cut.

The approach of this paper in not fully explained with the technical and theoretical details, more details need to be explained.

Experimental design

• Equation #7 should be Sj0 instead of Si0.

• Line 296: The gradient-associating sensors is not identified anywhere earlier in the paper.

• Equations should be after the parameters are identified for better understanding.

• Why the number of the simulation runs is 10? No explanation of why this value is chosen, in the literature, values converged after 45 runs.

Validity of the findings

• In all tables, the increase in values gradually with the increase in the number of sensors ; this is intuitively expected and should not be mentioned as a result. However results of optimizations are valid.

---

## Round 0.2 · Minor Revisions

The paper needs further improvements - the paper methodology should be clarified more. The authentication process should be explained as a flowchart figure.

Also the comments from the reviewers should be considered into account.

Reviewer 2 ·

Basic reporting

All comment seems to be resolved except the following:

• There is a need to explain the authentication problem in more details, some of the IIoT networks are designed in a closed network where they have their own cables and devices and no one outside the network has any access to it, in this case the authentication problem needs to be clarified at what circumstances it is considered an issue. I suggest the authors provide an example to explain when/how the authentication step should be included in the design.

• Authors need to identify the terminologies earlier in the paper such as authentication rate, authentication acceptance false rate, and authentication time.


I think authors should submit a corrected copy that marked all text change in red, it will be easier to see the new updates.

Experimental design

343-347 references and citations are required.

Validity of the findings

Conclusion is not sufficient, more details should be illustrated in this section.

---

## Round 0.3 · accepted · Accept

The revised paper is interesting and good to accept.

- A comprehensive and clearer conclusion is included.
- The paper has proper analysis can be shown to prove the strength of the approach.
- The description of the technical details is fine and this lead to improve the quality.
- The references in this manuscript are recent and Included the recent researches in this field.

Reviewer 2 ·

Basic reporting

All comments are resolved.

Experimental design

All comments are resolved.

Validity of the findings

All comments are resolved.

Additional comments

All comments are resolved.